# Photometric Determination of Iron in Pharmaceutical Formulations Using Double-Beam Direct Injection Flow Detector

**DOI:** 10.3390/molecules26154498

**Published:** 2021-07-26

**Authors:** Stanislawa Koronkiewicz

**Affiliations:** Department of Chemistry, University of Warmia and Mazury in Olsztyn, 10-957 Olsztyn, Poland; stankor@uwm.edu.pl; Tel.: +48-89-5234137

**Keywords:** pharmaceutical analysis, iron determination, spectrophotometry, flow analysis, direct injection detector, multi-pumping flow system

## Abstract

In this work, an innovative, flow-through, double-beam, photometric detector with direct injection of the reagents (double-DID) was used for the first time for the determination of iron in pharmaceuticals. For stable measurement of the absorbance, double paired emission-detection LED diodes and a log ratio precision amplifier have been applied. The detector was integrated with the system of solenoid micro-pumps. The micro-pumps helped to reduce the number of reagents used and are responsible for precise solution dispensing and propelling. The flow system is characterized by a high level of automation. The total iron was determined as a Fe(II) with photometric detection using 1,10-phenanthroline as a complexing agent. The optimum conditions of the propose analytical procedure were established and the method was validated. The calibration graph was linear in the range of 1 to 30 mg L^−1^. The limit of detection (LOD) was 0.5 mg L^−1^. The throughput of the method was 90 samples/hour. The repeatability of the method expressed as the relative standard deviation (R.S.D.) was 2% (n = 10). The method was characterized by very low consumption of reagents and samples (20 μL each) and a small amount of waste produced (about 540 µL per analysis). The proposed flow method was successfully applied for determination of iron in pharmaceutical products. The results were in good agreement with those obtained using the manual UV-Vis spectrophotometry and with values claimed by the manufacturers. The flow system worked very stably and was insensitive to bubbles appearing in the system.

## 1. Introduction

Iron is an essential nutrient in the human diet. It plays an important role in cellular processes such as respiration, synthesis of DNA and RNA, electron transport, and regulation of gene expression. Iron deficiencies leading to anemia are one of the world’s most common nutritional diseases. This is a global-scale public health problem affecting about 15% of human population (mainly children, teenagers, and woman at reproductive age) [1]. In order to avoid such deficiencies, an adequate supply of iron is needed. Iron preparations are one of the most commonly used medicaments.

Quality control of pharmaceuticals is a key factor at all stages of their development and production. The international pharmacopoeia requirements and standards related to quality control of pharmaceuticals have become increasingly strict. Most methods used for the assay of iron in pharmaceuticals are based on spectrophotometry [2,3,4,5]. Several other methods have also been reported including atomic absorption spectrometry [6], chemiluminescence [7], and electrochemical methods [8,9]. Most of these methods are easily applicable as detection methods in automated flow analysis systems. In such configuration, the most popular one is again spectrophotometric detection [3,4,5,10,11] due to its versatility, simplicity, rapidity, and relatively low cost.

For dedicated analytical applications, highly advanced and expensive spectrophotometers are unnecessary. Simple optical devices based on semiconductor components can be sufficient [12,13,14,15]. Spectrophotometric detection in such devices is achieved using light emission diodes (LEDs) as a source of inexpensive, stable, nearly monochromatic and high-intensity light. They are increasingly popular as a part of optical sensors since they offer several advantages including small size, low power consumption, robustness and long lifespan. In order to detect the light, the LEDs are usually coupled with a photodiode [16] or a phototransistor [10]. Optical detectors operating according to the paired emitter-detector diode (PEDD) principle are also commonly constructed [13,17,18,19,20]. PEDDs utilize two ordinary LEDs, the first of which is used as a source of the light and the second of which serves as a partially selective light detector. Wavelength selectors and optical fiber are unnecessary, allowing for reduction of optical complexity. Appropriately designed PEDDs have been successfully applied for photometric analytical measurements of iron in pharmaceuticals [21], as well as in wastewater and wine samples [11].

Application of flow analysis techniques has created new possibilities for automation of analytical procedures, often improving the precision, reducing the consumption of reagent, and increasing the sample throughput. Waste production and cost of analysis are reduced. Unfortunately, commercially available devices mostly rely on technology dating back whole decades, mainly continuous flow analysis (CFA) and flow injection analysis (FIA). More advanced techniques, such as sequential injection analysis (SIA) and systems commonly called “Lab-on-valve” (LOV, also called micro-SIA), are also available [4]. Modern solutions, better suited to the requirements of contemporary analytical chemistry, have been developed using electronically controlled actuators. These include, among others, micropumps and multi-way valves controlled using a solenoid. Solenoid micropumps are used in a technique called MPFS (multi pumping flow system) [22,23]. Solenoid micropumps can replace both the peristaltic pump and the system for precise dosing of the sample and reagents. The micropumps are characterized by their small size, which allows for miniaturization of the entire flow system. They have high precision and accuracy of dosing comparable with that of micropipettes. Operation of those elements can be programmed with a computer.

MPFS systems can be coupled with an original direct injection detector (DID). This type of detector is based on a PEDD system and allows for direct injection of reagents into the photometric detector’s chamber without using a reaction coil. It utilizes the fact that reagents are effectively mixed when rapidly injected in counter-current [24]. As a result, the samples are mixed with the reagents within a fraction of a second, directly in the detector chamber. It allows the time of analysis to be considerably shortened and avoids unnecessary sample dilution. So far, two types of photometric PEDD-based DID detectors have been developed and described, namely a single-beam DID [18] and a double-beam DID [19]. The single-beam DID was successfully adapted to determine Fe(III) in ground water using the thiocyanate method [20] and to determine Fe(II) in wastewater and wine [11]. In the double-beam DID, a log ration amplifier to measure absorbance has been applied providing a voltage signal proportional to the absorbance (linear function). The double-beam DID is superior to the single-beam DID in terms of better reproducibility of the results due to its autozeroing function. The design of this system minimizes the Schlieren effect. The flow system is stable even over a long period of time and insusceptible to gas bubbles.

In pharmaceutical products, iron is in the Fe(II) oxidation state since this is the form of iron easily absorbed in human body. Moreover, the majority of pharmaceuticals contain antioxidants in excessive amounts compared to the concentration of Fe(II), and this prevents quantitative oxidation of Fe(II) to Fe(III). Sample preparation is therefore difficult when oxidation of Fe(II) is needed before its determination as Fe(III). Sometimes it takes several hours to prepare the sample by digestion and separation of organic parts, and the process is very reagent-consuming [5]. Due to this, a direct, simple, precise and accurate, robust, and inexpensive method for determination of Fe(II) in pharmaceutical products would be greatly desired.

This work describes a novel, simple, automated procedure for iron determination in pharmaceuticals. In this method total iron was determined as Fe(II) through photometric detection using 1,10-phenanthroline as a complexing agent. In order to achieve correct analytical parameters, an automated method using a double-beam DID system was used. This is the first time this detector was applied for analytical purposes in pharmaceutical analysis. The full analytical features of the method were investigated.

## 2. Results and Discussion

### 2.1. Optimization of the Parameters

#### 2.1.1. Stop-Flow Time

For Fe(II) spectrophotometric determination of iron, the selective method based on the reaction between iron(II) and 1,10-phenanthroline is recommended (Scheme 1):

This reaction is known to be relatively slow. The time required for complete formation of the reaction product (ferroin complex) and recommended for batch condition is about 5 min [25]. Application of direct injection detectors allows for the colored product of the reaction to be monitored from the moment of mixing appropriate reagents [18,19,20]. Once that time (i.e., 5 min) has passed, the signal of absorbance should be stable. In order to verify whether this was true in the described method and to determine the optimum stop-flow time, the change of absorbance during the 300 s was registered. It was found that the absorbance signal is stable after about 90 s (Figure 1). However, in reproducible flow conditions the time required to read the analytical signal can be significantly shortened. Therefore, as the minimum stop-flow time, 30 s can be selected. This is a compromise between the time of analysis and the height of the analytical signal. In order to obtain the highest possible signal, a stop-flow time of 90 s is recommended. Most of the experiments presented in this paper were conducted using a stop-flow time of 70 s The choice of a stop-flow time of 70 s allowed to obtain high analytical signal in a relatively short time.

#### 2.1.2. Chemical Parameters

The effect of various chemical parameters was evaluated following the univariate method. At the beginning the pH of carrier solution was optimized. The studies were conducted in the range from 0 to 13 using appropriate concentration of NaOH or HCl (Figure 2). It was observed that the absorbance was approximately on the same level for pH in the range from 2 to 12. Therefore, the deionized water was chosen as a carrier. This facilitates saving the reagents. All standard solutions and samples were prepared in HCl of concentration 1.2 mol L^−1^ because the reduction of iron(III) to iron(II) using ascorbic acid has to be carried out under very acidic condition [25,26].

The concentration of 1,10-phenanthroline was studied between 0.1 and 1.2%. It was observed that the analytical signal steadily increased with increasing concentration (Figure 2b). However, for the concentration of 0.6%, the precision (standard deviation) was found the best and this concentration was chosen as a compromise between quality of the analytical signal and cost of analysis.

When formation of ferroin complex was realized in flow conditions, a solution of 1,10-phenentroline in sodium acetate was applied [26,27]. The concentration of CH_3_COONa in 1,10-phenanthroline solution was studied in the range of 0.1 to 1.5 mol L^−1^ (Figure 2c). It was observed that the absorbance was very low when the concentration of sodium acetate was below 0.6 mol L^−1^. Then the absorbance increased and was stable for concentrations higher than 1 mol L^−1^. Unfortunately, over the sodium acetate concentration of 1 mol L^−1^, an increase in blank signal was observed. Therefore, the concentration of 1 mol L^−1^ was chosen as an optimum. The level of blank in all experiments conducted under optimized conditions was always stable and very low, below 0.01.

### 2.2. Method Evaluation

#### 2.2.1. Analytical Parameters

The proposed flow system was evaluated while maintaining the optimum conditions described above. Linear calibration curves were attained for Fe(II) concentration between 1 and 30 mg L^−1^. The typically obtained relationship between absorbance and Fe(II) concentration is shown in Figure 3.

The limit of detection (LOD), calculated as 3*sb/S*, where *sb* is the standard deviation for 10 measurements of the blank and *S* is the slope of the calibration graph, was in the range of 0.5 mg L^−1^. Limit of quantification (LOQ), calculated as 10*sb/S*, was equal to 1.7 mg L^−1^. Taking into account the high content of iron(II) in the pharmaceuticals, the LOD value of the developed method is satisfactory.

The precision of the method evaluated as a relative standard deviation (R.S.D.) of 10 successive determinations of the standard solution 15 mg L^−1^ was found to be 2%. The estimated reagent consumption was very low, namely 20 µL of the sample and 20 µL of the 1,10-phenanthroline solution. The total volume of generated waste was 540 µL per analysis. Time of analysis applied for the calibration graph was 80 s (sample throughput of 45 determinations per hour). It can be shortened to about 40 s if very high sensitivity is unnecessary. In such condition the high throughput of 90 determinations per hour is possible.

#### 2.2.2. Application to the Real Samples

Usefulness of the presented flow system was evaluated through determination of iron(II) in a set of pharmaceutical formulations. The obtained results were compared with the results received from a batch manual spectrophotometric procedure and with a value claimed by the producer (information given on the packaging). The samples were diluted to obtain the concentrations in the range of calibration graph and next, the iron content was appropriately recalculated. The results are shown in Table 1.

To establish whether the proposed method produces reliable results and whether those results are in agreement with the traditional method of determination, Student’s t-test was applied. It was found that the calculated t value (from 0.20 for Sorbifer Durules to 0.96 for Hemofer Prolongatum) was lower than the tabulated t value (t = 2.78, n = 4, *p* = 0.05). This suggested that, at 95% confidence level, the difference between the results obtained by the proposed method and the reference method was statistically insignificant.

#### 2.2.3. Recovery Test

In order to check the accuracy of the proposed method, a recovery study was carried out. The samples of pharmaceutical formulations were spiked. The percentage of recoveries was calculated. It was in the range of 93 to 107%. The results shown in Table 2 confirm the validity of the method proposed.

### 2.3. Comparison of Proposed Methodology with Other Modern Metodologies

Comparison of the analytical features of the proposed approach with other systems, e.g., traditional, stationary spectrophotometry [2], flow methods applying spectrophotometers [3,4,5] and a commercially available flow system [4] was carried out (Table 3). The detection methods described in previously mentioned publications mainly concern conventional spectrophotometers [2,3,4,5] and LED-based techniques [10,11]. Various types of flow methods were also used for comparison: FIA [3], SIA [4,5], multicommutation based on solenoid valve application [10] and MPFS [11]. The detection process based mainly on the reaction of Fe(II) with 1,10-phenanthroline.

It can be concluded that the discussed method is distinguished from others by its high repeatability (R.S.D. of about 2%) and wide working range (from 0.5 to 30 mg L^−1^), which is satisfactory when one considers iron(II) content in pharmaceuticals. Limit od detection (LOD) and limit of quantification (LOQ) depend mainly on analytical reaction applied. For methods using 1,10-phenanthroline LOD (or LOQ) is similar. The differences may arise from different methods of determining this value.

The sampling rate (method’s throughput) is also very good, comparable to the commercially available system FIAlab [4]. Presented methodology allows for very low sample and reagent consumption (both of 20 µL) and low waste generation (total volume of about 540 µL per analysis). Generally, all flow methods should be expected to use less reagents than stationary, classical methods [2]. However, typical flow systems (e.g., FIA, SIA) [3,4,5,10] require the use of peristaltic pumps and reaction coils. As a result, the volumes of the solutions used are still relatively large. Flow system with DID detector, described in this work, do not use such elements. This allows for an even greater reduction in sample/reagent consumption and waste generation than FIA and SIA.

Comparing the obtained results with those of the simplified, single-beam version of the DID [11], it can be concluded that the analytical parameters have been significantly improved.

## 3. Materials and Methods

### 3.1. Chemicals and Reagents

All solutions were prepared with analytical-grade chemicals and using deionized water obtained from a Milli-Q (Millipore) water purification system (resistivity > 18.2 MΩ cm). A 1 mg mL -1 stock solution of Fe(II) was prepared by dissolving 0.4982 g of iron(II) sulfate heptahydrate (Sigma-Aldrich, Darmstadt, Germany) in 100 mL of 1.2 mol L^−1^ HCl (Merck, Darmstadt, Germany). Working standard solutions were prepared by appropriate dilution of stock solution with 1.2 M of HCl to obtain the final concentration of Fe(II) ions from 1 to 30 mg L^−1^. All working solutions additionally contained a reducing agent, namely ascorbic acid (Chempur, Poland) at a concentration of 10 mg L^−1^. Solutions of ascorbic acid were prepared daily. The chromogenic reagent (solution of 1,10-phenanthroline) was prepared by dissolving an appropriate amount of 1,10-phenanthroline monohydrate (Sigma-Aldrich, Germany) in a mixture of ethanol and sodium acetate (5:25, *v*/*v*) (POCh, Poland). As a carrier, solutions of HCl, NaOH (POCh, Poland) or deionized water were used.

### 3.2. Samples

For the demonstration of practical utility of the developed flow system, samples of pharmaceutical formulations listed in Table 4 were used.

All pharmaceuticals used also contained different additives (e.g., talc, stearic acid, beeswax, paraffin, microcrystalline cellulose) which were insoluble in water. To prepare the samples, the tablets or the contents of capsules/sachets were placed into the mortar and carefully grinded. Then, the resulting powder was dissolved in 100 mL of 1.2 mol L^−1^ HCl with addition of 10 mg L^−1^ of ascorbic acid to prevent oxidation of iron(II) to iron(III). Insoluble pharmaceutical ingredients were filtered. If necessary, the extracts were additionally diluted 10- to 100-fold using HCl and ascorbic acid before analysis to obtain the iron content compatible with the linear range of determination. The pharmaceuticals also contained some water-soluble excipients which were presented in the sample matrix. The most important and common were: ascorbic acid, lactose, sucrose, gelatin, acacia gum, and maltodextrin. Due to the highly selective nature of the reaction of Fe(II) ions with 1,10-phenanthroline, it was assumed that they should not significantly affect the analytical signal.

### 3.3. Apparatus

#### 3.3.1. Flow System

The flow manifold consisted of three solenoid-operated micro-pumps, flow lines, and a double-beam direct-injection detector (double-DID). The micro-pumps were purchased from Cole-Parmer (Boonton, USA) and have a nominal volume of 20 µL (product no. P/N 73120-10) or 50 µL (product no. P/N 73120-22). The flow lines were made of a PTFE tube (ID 0.8 mm) and were also obtained from Cole-Parmer. The 20 µL volume pumps were used for injecting the sample and the chromogenic reagent into the reaction chamber of the DID. The third micro-pump was used for propelling the carrier. To decrease the time of analysis, the nominal volume of this pump was bigger (50 µL). The schematic diagram of the applying flow network is shown in Figure 4.

The double-beam DID was made using one block of Teflon. Inside the detector there were two cells identical in shape and size, reaction (RC) and reference (RF). The optical path length was 20 mm, and the volume of each of the cells was 60 µL. All details regarding the principle of operation and characteristics of this detector were discussed in an earlier publication [19]. As emission diodes for iron(II) determination, two identical green LEDs were chosen. The maximum of the emission spectrum of these LEDs was consistent with the maximum of absorption of the Fe(II)-1,10-phenanthroline (ferroin) complex (λ_max_ = 512 nm [25]). Since the spectral sensitivity of the LEDs is usually shifted towards the shorter wavelength (Stokes shift) [13,14,15], two yellow-green LEDs with emission λ_max_ = 540 nm were selected as detection diodes, since they were well suited to detection of light emitted by the green diodes, which have lower λ_max_. The LEDs were purchased at a local electronics parts shop.

The work of the DID and the micro-pumps were PC-controlled by the measurement system developed specifically for flow analysis in our laboratory [28]. The software enables the user to control the work of the solenoid micro-pumps and to calibrate the signal of absorbance. For absorbance measurements, an integrated logarithmic amplifier LOG101 (Burr-Brown, Tucson, AZ, USA) was used.

#### 3.3.2. Procedure

All solutions were aspirated and then properly injected into the detector by three independent solenoid micro-pumps: P1, P2, and P3. The pumps were responsible not only for the propelling of all solutions, but also for precise dispensing. Rapid injection in counter-current facilitated efficient mixing of the solutions used. An example of the micro-pumps program used for the Fe(II) determination is shown in Figure 5.

Before the analytical cycle began, both cells of the detector (reaction (RC) and reference (RF)) were filled with the carrier solution using pump P3. The measuring cycle started by zeroing the absorbance signal (Figure 5, section A).

Next, simultaneous injection of the sample (pump P1) and the solution of 1,10-phenanthroline (pump P2) occurred (Figure 5, section B). Injection was rapid and conducted in counter-current. Such a method of introduction facilitated effective homogenization of the reagents. That way the reaction cell was filled with the total volume of sample and reagent of 40 µL. This volume was lower than the volume of the cell (60 µL) and the absorbing the light molecules should not escape the reaction cell.

From the moment of sample and reagent injection and mixing, the reaction started and absorbance was measured (Figure 5, section C). The stop-flow time for this section and, consequently, the measuring cycle time, depended mainly on the kinetics of the reaction used.

At the end of each cycle, the detector was cleaned using the carrier, by pump P3 (Figure 5, section D). To ensure the removal of reagents from reaction cell was thorough, several strokes of pump P3 were applied. The influence of the contaminants originating from previous cycle was not observed when the detector was cleaned using about 500 µL of the carrier.

#### 3.3.3. Reference Method

The manual procedure for determination of Fe(II) used was identical to that recommended by Marczenko [25]. Absorbance was measured at 512 nm with a conventional spectrophotometer (Tomos Life Science Group, Singapore, model V-1100).

## 4. Conclusions

It can be concluded that the proposed novel flow procedure for iron(II) determination is a very valuable solution from the point of view of quality control analysis of pharmaceuticals. The chief advantages of the presented flow system are its very high stability and repeatability, high sampling frequency, simple and compact construction and high level of automation. Obtained results confirmed the analytical usefulness of the developed approach, and they are highly repeatable and accurate. Additionally, the method is inexpensive and environmentally friendly, since the reagent consumption and total waste production are very low (in the range of 20 µL and 540 µL per analysis, respectively).

The described system can work independently. However, it is possible to combine it with other flow systems, resulting in a more sophisticated overall system. It would be worthwhile to conduct research in the direction of developing a procedure for on-line (in a flow system) sample preparation. Obviously, this would increase the degree of automation of the entire determination, and would allow for a real, final minimization of sample consumption. In such configuration described flow system can be especially recommended for the Green Chemistry monitoring station working in a fully automatic mode.

## 5. Patents

The technical solutions and potential applications in analytical chemistry of a double-beam direct injection detector are described in Polish Patent [29].

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
