# Peer review of "Photometric Determination of Iron in Pharmaceutical Formulations Using Double-Beam Direct Injection Flow Detector"

_molecules, 2021, doi:10.3390/molecules26154498_

Round 1

Reviewer 1 Report

In this article, Koronkiewicz present a novel flow procedure for the iron determination in pharmaceuticals. The method is clearly explained and the perfomance is compared with a traditionatal procedure. For these reasons, we consider that this article deserves to be published in Molecules with minor revisions.

The title suggests that the method is limited to antianemic pharmaceutical formulations, but in the article it seems that it can be applied to all kind of pharmaceutical products. Then, the title should be reformulated accordingly.

The equations of the reactions involved in the process should be included.

In line 180, the author states "The obtained results were compared with the results received from a batch manual spectrophotometric procedure and with a value claimed by the producer."

The reference is missing.

In line 295, the author states: "Injection was rapid and done in counter-current. Such a method of introduction facilitated effective homogenization of the reagents."

The last sentence is quite obvious.

The author mentions the low waste generation of this methodology, which seems to be right, but there is no reference about this point for other methods

The quality of Figures 2 and 3 should be improved.

Some perspectives and future work should be mentioned.

Reviewer 2 Report

The manuscript contains interesting work on the flow injection analysis, the modern way of Fe(II) analysis. It is well written. I have just minor comments:

1) Line 121; author descibed that the signal is stable with 90 s, 30 s is the compromise, but "most of the experiments ... were conducted using 70 s". Why?

2) Line 162; LOQ should be also given. LOD or LOQ values should be also given in the Table 3 to compare the methodologies.

3) Line 221 + 247; author stated that the methodology is "low-consumption" and "low-waste"; however, the pharmaceutical formulations were dissolved in 100 mL. This is not in the correlation with the information about the low volume of the waste. Is it necessary to use 100 mL for the extraction?

4) Author stated "no conflict of interest" but also gave the link for the patent. Is the "conflict of interest" correct?

Reviewer 3 Report

  1. This paper measures the concentration of iron (II) in antianemic pharmaceutical formulation, what are the specific characteristics of the matrices of aforementioned drugs.
  2. Fe2+ is quantitatively complexed by 1,10-phenanthroline in the pH range from 3 to 9. Sodium acetate is used as a buffer to maintain a constant pH at 3.5. If the pH is too high, the Fe2+ will be oxidized to Fe3+; if the pH is too low, H+ will compete with Fe2+ for the basic 1,10-phenanthroline (to form phenH+). However, your paper shows different result. What is the reason for that.
  3. What are the originality and advantages of this method?

Round 2

Reviewer 3 Report

This manuscript can be accepted as current form.